# Assessment of Liver Fibrosis Stage Using Integrative Analysis of Hepatic Heterogeneity and Nodularity in Routine MRI with FIB-4 Index as Reference Standard

**DOI:** 10.3390/jcm10081697

**Published:** 2021-04-15

**Authors:** Tae-Hoon Kim, Chang-Won Jeong, Ji Eon Kim, Jin Woong Kim, Hoon Gil Jo, Youe Ree Kim, Young Hwan Lee, Kwon-Ha Yoon

**Affiliations:** 1Medical Convergence Research Center, Wonkwang University, Iksan 54538, Korea; mediblue@wku.ac.kr (C.-W.J.); kakasky112@wku.ac.kr (J.E.K.); 2Smart Health IT Center, Wonkwang University Hospital, Iksan 54538, Korea; 3Department of Radiology, Chosun University College of Medicine, Chosun University Hospital, Gwangju 61452, Korea; jw4249@hanmail.net; 4Department of Hepatology & Gastroenterology, Wonkwang University Hospital, Iksan 54538, Korea; jojo4206@hanmail.net; 5Department of Radiology, Wonkwang University School of Medicine, Wonkwang University Hospital, Iksan 54538, Korea; sweetynn@naver.com (Y.R.K.); yjyh@wku.ac.kr (Y.H.L.)

**Keywords:** chronic liver disease (CLD), heterogeneity, nodularity, integrative

## Abstract

Image-based quantitative methods for liver heterogeneity (L_Het_) and nodularity (L_Nod_) provide helpful information for evaluating liver fibrosis; however, their combinations are not fully understood in liver diseases. We developed an integrated software for assessing L_Het_ and L_Nod_ and compared L_Het_ and L_Nod_ according to fibrosis stages in chronic liver disease (CLD). Overall, 111 CLD patients and 16 subjects with suspected liver disease who underwent liver biopsy were enrolled. The procedures for quantifying L_Het_ and L_Nod_ were bias correction, contour detection, liver segmentation, and L_Het_ and L_Nod_ measurements. L_Het_ and L_Nod_ scores among fibrosis stages (F0–F3) were compared using ANOVA with Tukey’s test. Diagnostic accuracy was determined by calculating the area under the receiver operating characteristics (AUROC) curve. The mean L_Het_ scores of F0, F1, F2, and F3 were 3.49 ± 0.34, 5.52 ± 0.88, 6.80 ± 0.97, and 7.56 ± 1.79, respectively (*p* < 0.001). The mean L_Nod_ scores of F0, F1, F2, and F3 were 0.84 ± 0.06, 0.91 ± 0.04, 1.09 ± 0.08, and 1.15 ± 0.14, respectively (*p* < 0.001). The combined L_Het_ × L_Nod_ scores of F0, F1, F2, and F3 were 2.96 ± 0.46, 5.01 ± 0.91, 7.30 ± 0.89, and 8.48 ± 1.34, respectively (*p* < 0.001). The AUROCs of L_Het_, L_Nod_, and L_Het_ × L_Nod_ for differentiating F1 vs. F2 and F2 vs. F3 were 0.845, 0.958, and 0.954; and 0.619, 0.689, and 0.761, respectively. The combination of L_Het_ and L_Nod_ scores derived from routine MR images allows better differential diagnosis of fibrosis subgroups in CLD.

## 1. Introduction

Liver fibrosis is a hallmark of chronic liver disease (CLD) characterized by excessive accumulation of extracellular matrix proteins responsible for fibrogenesis [1,2]. Liver fibrosis may progress to cirrhosis, the end stage, which constitutes the most important risk factor for developing hepatocellular carcinoma (HCC) [3]. Liver biopsy has been regarded as the reference diagnostic method for evaluating the stage of liver fibrosis in CLD [4]. However, this method has well-known weaknesses including sampling errors, low patient acceptance, and complications such as pain, bleeding, infection, and rarely death [5]. Moreover, in the CLD patients with an initial diagnosis of early stage fibrosis (compensated liver) or cirrhosis (decompensated liver as end stage), it is difficult to accurately predict hepatic compensation or decompensation using noninvasive methods [6]. Thus, there is an unmet need for widely applicable noninvasive methods to diagnose fibrosis and advanced cirrhosis and to predict future risk of hepatic decompensation.

Recently, there have been considerable efforts to develop imaging techniques and quantification programs for diagnosing and staging liver fibrosis. There are several methods including contrast-enhanced imaging, elastography, image-based morphologic analysis, and texture analysis [7,8,9]. Among them, image-based morphologic analysis includes quantification of parenchymal heterogeneity, contour change by the liver nodules, atrophic or necrotic change, edge blunting, fissural widening, and so on [10]. Several quantification software program have been introduced for assessing the findings of liver fibrosis and cirrhosis on medical images [10,11,12]. Heterogeneity quantification programs using coefficient of variation (CV) maps can help to assess the severity of fibrosis and cirrhosis in patients with chronic hepatitis B [10,11]. Several studies [10,11] reported that the area under the receiver operating characteristic curves (AUROCs) on the magnetic resonance (MR) CV map scores were 0.875 for discriminating significant fibrosis (≥fibrosis grade 2; F2) in chronic hepatitis B and 0.788 for the presence of HCC in patients with liver cirrhosis (F4). Moreover, liver surface nodularity (LSN) can be useful for differentiating the severity of fibrosis. A study [12] reported that the AUROC was 0.788 for discriminating significant fibrosis (≥F2) in nonalcoholic fatty liver disease (NAFLD). Other studies showed that the AUROCs of computed tomography (CT) LSN scores were 0.902 and 0.959 for discriminating significant fibrosis (≥F2) and cirrhosis (F4), respectively [13,14]. A comparative study using MR LSN and MR elastography demonstrated that the AUROCs for diagnosing significant fibrosis (≥F2) were 0.61 for MR LSN and 0.87 for MR elastography [15]. However, these studies used a single measurement, either liver heterogeneity (L_Het_) or nodularity (L_Nod_), for evaluating liver fibrosis; the method was not clearly distinguished among each fibrosis stage. Taking all of these findings into consideration, the computer-aided L_Het_ and L_Nod_ scores can provide important information for differential diagnosis of hepatic fibrosis. The integrative analysis of L_Het_ or L_Nod_ for evaluating liver fibrosis is not yet fully understood.

For this study, we developed an integrated semiautomated quantification software for assessing L_Het_ and L_Nod_ and compared them across fibrosis stages in CLD.

## 2. Materials and Methods

### 2.1. Ethics Statment

The study protocol was approved as retrospective research (WKUH-2017-03-026) by the institutional review board (IRB) of University Hospital. Written informed consent was waived by the University Hospital IRB committee due to the use of anonymous archival data including MRI data (radiology_common data model: R_CDM, version 2.0.0) and electronic health records (observational medical outcomes partnership-CDM: OMOP-CDM, version 5.3) for the application of developed software. This study was conducted in accordance with the Helsinki Declaration and Good Clinical Practice guidelines.

### 2.2. Subject Population

Among the 1654 consecutive patients who underwent radiological examination at our institution from April 2003 to December 2018, patients 20 years or older who underwent abdominal MRI at 3.0 T and who had available serologic tests within five months of MRI were retrospectively identified. Of 121 eligible patients, 10 were excluded due to the absence of MR images for liver protocols and the absence of medical records for CLD (Figure 1). The inclusion criteria of CLD patients were the elevation of liver function enzymes, alanine transaminase (ALT), and aspartate transaminase (AST) and the absence of liver cirrhosis (F4) [14]. The CLD subgroups were divided into three fibrosis groups according to the serum biomarkers of fibrosis-4 index (FIB-4, Equation 1) values as follows: F1, mild fibrosis group < 1.45; F2, significant fibrosis group 1.45–3.25; and F3, advanced fibrosis group > 3.25 (Table 1). Finally, the subgroups consisted of 9 F1 (mean age; 50.3 ± 14.9 years), 57 F2 (60.3 ± 12.1 years), and 45 F3 (mean 64.8 ± 13.6 years). This study included 16 subjects (35.0 ± 15.5 years) with suspected liver disease who underwent the needle biopsy for comparison (Figure 1). These individuals had symptoms of fatigue and inactivity. They had abnormal liver function tests, and there was no histological evidence for liver fibrosis and advanced cirrhosis (no fibrosis group, F0).
(1)$$\mathrm{FIB}-4=\frac{\left(\mathrm{Age}\left(\mathrm{year}\right)\times \;\mathrm{AST}\left(U/L\right)\right)}{\left(\mathrm{platelet}\;\mathrm{count}\;\left(10^{9}/L\right)\times \;\mathrm{square}\;\mathrm{root}\;\mathrm{of}\;\mathrm{ALT}\;\left(U/L\right)\right)}$$

The upper limit of normal AST was 35 in this study.

### 2.3. Acquisition of MRI

All MRI examinations were performed on a 3T MRI scanner (Achieva; Philips, Netherlands) with a 32-channel receiver body matrix coil. The T1WI were acquired with three-dimensional T1 high-resolution isotropic volume excitation (THRIVE) pulse sequence: TR/TE = 4.2/1.97 msec; field of view (FOV) = 38 × 38 × 14 cm^3^, matrix size = 512 × 512, number of excitation (NEX) = 2, slice thickness = 0.74 × 0.74 × 2.0 mm^3^, and number of slices = 70.

### 2.4. Software for Quantification of Liver Heterogeneity and Nodularity

L_Het_ and L_Nod_ quantification software (customized software; named WALTS) was coded by MATLAB (MathWorks, Natick, Massachusetts). Wonkwang Abdomen and Liver Total Solution (WALTS) software is a customized semiautomated postprocessing program that operates on Windows platform (client version: XP or higher; Microsoft, Redmond, WA). We used WALTS to process the MR images in the DICOM (Digital Imaging and Communications in Medicine) format to generate the L_Het_ and L_Nod_ scores using a previously described procedure [13,14,16]. Figure 2 shows the GUI of WALTS and a simple flowchart showing the development of an algorithm for qualitative and quantitative analysis. The procedures for quantifying L_Het_ and L_Nod_ scores were as follows: bias correction of field uniformity, liver contour detection for drawing the liver reference line, liver segmentation, and L_Het_ and L_Nod_ measurements.

### 2.5. Data Processing and Quantification of MRI in CLD

Figure 3 shows the overall image postprocessing procedures for hepatic heterogeneity (L_Het_) and nodularity (L_Nod_) quantification using MR images. To automatically detect the liver’s contour, we used a novel region-based method for liver segmentation as a level set method, which provided the local clustering criterion function with correction with intensity inhomogeneities (Figure 4B) [17]. The boundary detection and segmentation techniques maximize the local intensity clustering property and minimize the energy formulation to determine and exclude any existing signal outliers caused by generated systematic artifacts as described in previous studies [12,18]. The contour line in the selected slice of the liver was produced after bias correction. Following preprocessing of MRI data, the liver surface line for L_Het_ and L_Nod_ quantification was extracted as a reference line, and the extracted line was confirmed by two abdominal radiologists (with 29 and 8 years of experience in abdominal imaging) (Figure 4C). Five circular regions of interest (ROIs; each 40 pixels) for L_Het_ measurement were drawn on the liver parenchyma. In all subjects, ROIs were placed on the liver parenchymal areas with no overlap over large vessels or focal lesions. The L_Het_ score and L_Het_ map were calculated using the following Equations (2) and (3):(2)$${\;L}_{\mathrm{Het}}\;\mathrm{score}=\frac{\mathrm{Standard}\;\mathrm{Deviation}}{\mathrm{Mean}}\times 100$$
(3)$$L_{\mathrm{Het}}\;\mathrm{map}\;=\frac{\mathrm{CV}}{\mathrm{Pixel}\;\mathrm{Value}}\times 100$$

To measure the L_Nod_ score, the liver parenchyma within the confirmed liver boundary line was used for the multipolynomial curve fitting analysis. ROIs for L_Nod_ measurement were selected along the contour of the liver (Figure 4G). The user would insert an L_Nod_ ROI range across the data points of the liver surface line. After input of an L_Nod_ ROI range, a smooth curve-fitting line (polynomial line shape) was generated on a selected ROI dataset (Figure 4H). Finally, the difference between the liver surface line and the new polynomial curve-fitting line (one of second-, third-, and fourth-order line shape) was evaluated on a pixel-by-pixel basis. The difference value was squared; then it was used to calculate the mean, variation, and standard deviation (SD). The final L_Nod_ score in an individual subject was calculated as the mean L_Nod_ obtained from the measurements on ROIs. In addition, a combined score derived from L_Het_ and L_Nod_ was calculated as the multiplication of both scores (=L_Het_ × L_Nod_).

All MR studies reviewed standard picture archiving and communication system (PACS) stations and software with standard window settings. The liver MR images in each CLD patient were assessed by two abdominal radiologists, who were blinded to clinical outcome, using WALTS software. After opening DICOM images on the software, they selected image slices at the level of the hepatic hilum. Then, bias correction and segmentation were performed on the selected images. For L_Het_ measurement in each subject, five circular ROIs were manually drawn in the liver parenchyma (Figure 5); these areas contained the liver parenchyma, avoiding the perceivable bile duct, major intrahepatic vessels, subcapsular area, and focal lesions such as cysts or benign and malignant tumors [11]. The final L_Het_ and L_Nod_ scores in each fibrosis group were calculated as an averaged score obtained by reporting scores of each observer (observer A: YRK, observer B: YHL) for AUROC differential diagnosis according to fibrosis stages. 

All the measurements on selected MR images (the level of the hepatic hilum) were repeated two weeks after the first measurement was obtained to evaluate intraobserver agreement. Furthermore, to determine interobserver agreement, both radiologists independently measured L_Het_ and L_Nod_ scores on selected images. The intra- and interobserver variability in the L_Het_ and L_Nod_ measurements was assessed. The overall scores of L_Het_ and L_Nod_ were calculated as the mean score of the three measurements taken for each patient. The WALTS program for L_Het_ and L_Nod_ quantification used MR images in DICOM format to generate the L_Het_ and L_Nod_ scores. The technical details for obtaining the L_Het_ and L_Nod_ measurements were described in recent papers [10,11,12,13]. Measurements of at least three and/or four ROIs were performed for each subject. Final L_Het_ and L_Nod_ scores were calculated by the program as a mean value of the individual measurements, with a higher L_Het_ (or L_Nod_) score indicating a higher degree of parenchymal heterogeneity (or nodularity). Figure 4 shows the representative images in L_Het_ and L_Nod_ measurements in an axial MR image.

### 2.6. Statistical Analysis

The L_Het_ and L_Nod_ scores among three different stages of fibrosis in CLD were compared using the SPSS version 20.0 program (SPSS Inc., Chicago, IL, USA). The variation in L_Het_ and L_Nod_ scores was analyzed using analysis of variance (ANOVA) with Tukey’s post hoc test. The difference between CLD patients and the control group was analyzed using the independent two-sample *t*-test. Intraobserver agreement (between measurements from the same observer) was calculated as the mean coefficient of variance (%) for the variability of L_Het_ and L_Nod_ scores taken by the same single observer [19]. Also, the variation between the scores of both observers was analyzed with paired *t*-test. Intraobserver agreement was performed on the basis of the intraclass correlation coefficient (ICC) between the L_Het_ and L_Nod_ scores. The ICCs were indicated based on the levels of reliability as follows [20]: poor (<0.4), moderate (0.4–0.6), good (0.6–0.8), and excellent (0.8–1.0). 

The diagnostic performance of L_Het_, L_Nod_, and L_Het_ × L_Nod_ scores according to fibrosis stages was evaluated with ROC curve analysis including the AUROC, sensitivity, and specificity. Two-sided *p*-values less than 0.05 were considered to denote statistical significance in all tests.

## 3. Results

### 3.1. Patient Characteristics in Fibrosis Stages

Figure 1 shows the inclusion flowchart for the study population. The etiology of liver fibrosis in CLD and the average enzyme levels according to fibrosis stages are listed in Table 1. The serum biochemistry showed significant difference among three groups in the levels of alkaline phosphatase (ALP, *p* = 0.002), glutamyl transpeptidase (GGT, *p* = 0.001), and platelet count (*p* < 0.001). However, there was no significant difference among the fibrosis groups as follows: albumin (*p* = 0.873), alanine aminotransferase (ALT, *p* = 0.224), aspartate aminotransferase (AST, *p* = 0.058), and bilirubin (*p* = 0.058).

### 3.2. Liver Heterogeneity and Nodularity Measurements in CLD

MRI data for 111 CLD patients and 16 subjects with suspected liver disease were analyzed with developed WALTS software. Figure 3 shows the detailed image processing procedures. Figure 4 shows a representative MR image of a patient with CLD (Figure 4A), bias-corrected MR image (Figure 4B), liver contour detection (Figure 4C), region of interest (ROI) drawing on MR image (Figure 4D), binary image of ROI for liver segmentation (Figure 4E), segmented liver MRI (Figure 4F), final reference line (Figure 4G), and curve-fitting lines (Figure 4H) for L_Het_ and L_Nod_ measurements.

Figure 5 shows the representative quantification images in each fibrosis stage (F1–F3), and the results are summarized in Table 2. Mean L_Het_, L_Nod_, and L_Het_ × L_Nod_ scores in CLD were higher than those in the F0 group (*p* < 0.001). Mean L_Het_, L_Nod_, and combined L_Het_ × L_Nod_ scores were significantly different between fibrosis stages (F1–F3) (ANOVA; *p* < 0.001). In multiple comparisons, L_Het_ scores were different from each other (F1 vs. F2, *p* = 0.032; F1 vs. F3, *p* < 0.001; and F2 vs. F3, *p* = 0.008). Additionally, L_Nod_ scores were different in F1 vs. F2, *p* < 0.001; F1 vs. F3, *p* < 0.001; and F2 vs. F3, *p* = 0.022. The combined scores were significantly different in the contrasts of F1 vs. F2, F1 vs. F3, and F2 vs. F3 (*p* < 0.001). There were significant differences among fibrosis stages based on quantitative L_Het_, L_Nod_, and L_Het_ × L_Nod_ scores.

### 3.3. ROC Analysis for Differential Diagnosis According to Fibrosis Stages

Figure 6 shows AUROC curves of L_Het_, L_Nod_, and L_Het_ × L_Nod_ scores for the discrimination of fibrosis stages, and the results are summarized in Table 3. The AUROCs of L_Het_ scores were 0.845 for F1 vs. F2 (95%CI, 0.722–0.968; *p* = 0.001), 0.619 for F2 vs. F3 (95%CI, 0.509–0.730; *p* = 0.040), 0.944 for F0–1 vs. F2 (95%CI, 0.890–0.997; *p* < 0.001), and 0.963 for F0–1 vs. F3 (95%CI, 0.918–1.000; *p* < 0.001). The diagnostic accuracy of F1 vs. F2 had 0.714 sensitivity and 0.778 specificity at a cut-off L_Het_ score of 6.22; F2 vs. F3 had 0.533 sensitivity and 0.536 specificity at a cut-off L_Het_ score of 7.02; F0–1 vs. F2 had 0.875 sensitivity and 0.880 specificity at a cut-off L_Het_ score of 5.68; and F0–1 vs. F3 had 0.867 sensitivity and 0.880 specificity at a cut-off L_Het_ score of 6.17. 

The AUROCs of L_Nod_ scores were 0.958 for F1 vs. F2 (95%CI, 0.890–1.000; *p* < 0.001) and 0.689 for F2 vs. F3 (95%CI, 0.577–0.801; *p* = 0.001), 0.969 for F0–1 vs. F2 (95%CI, 0.931–1.000; *p* < 0.001) and 0.936 for F0–1 vs. F3 (95%CI, 0.872–0.999; *p* < 0.001). The diagnostic accuracy of F1 vs. F2 had 0.947 sensitivity and 1.000 specificity at a cut-off L_Nod_ score of 0.97; F2 vs. F3 had 0.644 sensitivity and 0.649 specificity at a cut-off L_Nod_ score of 1.12; F0–1 vs. F2 had 0.946 sensitivity and 0.960 specificity at a cut-off L_Het_ score of 0.95; and F0–1 vs. F3 had 0.911 sensitivity and 0.920 specificity at a cut-off L_Het_ score of 0.94. 

The AUROCs of L_Het_ × L_Nod_ scores were 0.954 for F1 vs. F2 (95%CI, 0.884–1.000; *p* < 0.001) and 0.761 for F2 vs. F3 (95%CI, 0.669–0.853; *p* < 0.001), 0.984 for F0–1 vs. F2 (95%CI, 0.960–1.000; *p* < 0.001) and 0.999 for F0–1 vs. F3 (95%CI, 0.996–1.000; *p* < 0.001). The diagnostic accuracy of F1 vs. F2 had 0.895 sensitivity and 0.889 specificity at a cut-off L_Het_ × L_Nod_ score of 6.10; F2 vs. F3 had 0.667 sensitivity and 0.684 specificity at a cut-off L_Het_ × L_Nod_ score of 7.85; F0–1 vs. F2 had 0.911 sensitivity and 0.920 specificity at a cut-off L_Het_ × L_Nod_ score of 5.98; and F0–1 vs. F3 had 0.978 sensitivity and 1.000 specificity at a cut-off L_Het_ × L_Nod_ score of 6.70.

Diagnostic accuracy (%, number of subject) of L_Het_ scores in discriminating the FIB-4 stages was 72.7% (48/66) for F1 vs. F2, 52.9% (54/102) for F2 vs. F3, 87.8% (72/82) for F0–1 vs. F2 and 87.1% (61/70) for F0–1 vs. F3. Diagnostic accuracy of L_Nod_ scores was 95.4% (63/66) for F1 vs. F2, 64.7% (66/102) for F2 vs. F3, 95.1% (78/82) for F0–1 vs. F2, and 91.4% (64/70) for F0–1 vs. F3. Diagnostic accuracy of L_Het_ × L_Nod_ scores was 89.4% (59/66) for F1 vs. F2, 67.6% (69/102) for F2 vs. F3, 91.5% (75/82) for F0–1 vs. F2, and 98.6% (69/70) for F0–1 vs. F3.

### 3.4. Intraobserver and Interobserver Agreement

The intra- and interobserver variabilities of L_Het_ and L_Nod_ scores from two observers are summarized in Table 4. In intraobserver variability, the mean coefficient of variation within the same observer was in the range of 13–28% for L_Het_ measurements and in the range of 4–14% for L_Nod_ measurements. Also, there was no significant difference between the averaged L_Het_ and L_Nod_ values of the two observers (*p* > 0.05). In interobserver variability, ICCs were higher than 0.6, indicating good reliability. The ICCs (range: 0.601–0.852) were 0.718 for L_Het_ measurements and 0.832 for L_Nod_ measurements. The overall L_Het_ and L_Nod_ measurements of both observers showed good agreement (*p* < 0.05).

## 4. Discussion

This study developed an integrated system (semiautomated WALTS software) for evaluating L_Het_ and L_Nod_ in liver diseases and compared the subgroups of fibrosis stages in CLD patients obtained from retrospective routine MRI datasets with serologic laboratory tests. In this study, liver MR images with three-dimensional THRIVE pulse sequence (routine T1 MR images) demonstrated acceptable accuracy in diagnosing fibrosis stages of CLD patients. L_Het_, L_Nod_, and L_Het_ × L_Nod_ scores in CLD patients were higher than those in the F0 group. The AUROC-based differentiation in comparison of F1 vs. F2 fibrosis was significant as L_Het_ 0.845, L_Nod_ 0.958, and L_Het_ × L_Nod_ 0.954. Moreover, the AUROC in F2 vs. F3 was significant as L_Het_ 0.619, L_Nod_ 0.689, and L_Het_ × L_Nod_ 0.761. Smith et al. [13] and Pickhardt et al. [14] reported that the L_Nod_ diagnostic accuracy using CT images is excellent for predicting fibrosis (≥F2) or cirrhosis (F4) (0.910 and 0.959 AUROC, respectively). Furthermore, Lee et al. [10] reported that the mean L_Het_ values showed good discrimination for staging of significant fibrosis (≥F2) in chronic hepatitis B (aspartate aminotransferase to platelet ratio index: APRI 0.875 and FIB-4 0.831 AUROC). In the present study, the L_Het_ and L_Nod_ scores in the CLD patients are in accordance with these previous results [10,11,13,14], confirming patients with significant fibrosis (≥F2) and/or precirrhotic hepatic fibrosis.

This study investigated the potential variation in L_Het_ and L_Nod_ measurements and interobserver assessment. To successfully detect signals from the liver parenchyma and surface, all T1-weighted MRI data were performed for bias correction of field homogeneity before the liver contour detection. Quantitative L_Het_ and L_Nod_ scores showed reliable measurements as an averaged CV value <25%. The L_Het_ and L_Nod_ scores measured from two observers showed good interobserver agreement (>0.6), indicating reproducibility. Thus, the WALTS software-based L_Het_ and L_Nod_ measurements can be reproducible in clinical MR images. However, the most accurate test for assessing liver fibrosis is currently MR elastography (MRE), which has ICC >95% and accuracy >90% in liver stiffness measurements [21]. In our study, the L_Het_ and L_Nod_ have ICC of 0.72 and 0.83, respectively, which is quite low compared to MRE. Thus, further study is needed for a more accurate quantification method in the L_Het_ and L_Nod_ measurements. 

With regards to the grading of liver fibrosis, the APRI and FIB-4 serologic indices are well known [22,23]. In a meta-analysis study [22], the pooled ROCs of the FIB-4 index were 0.74–0.84 in the patients with chronic hepatitis B virus infection. The summary ROC (SROC) values of FIB-4 were higher than those of APRI for advanced fibrosis and cirrhosis. Two systematic reviews [23,24] reported that the SROC values for the accuracy of APRI in patients with hepatitis C (HCV) or coinfection of HCV/human immunodeficiency virus (HIV) were 0.76–0.77 for significant fibrosis, 0.80 for advanced fibrosis, and 0.82–0.83 for cirrhosis. Based on these findings, the FIB-4 index for diagnosing liver fibrosis and cirrhosis has similar or superior diagnostic accuracy to that of APRI. For this study, we used the FIB-4 scoring system using serum ALT, AST, and platelet levels. The interesting features in this study are that the L_Het_, L_Nod_, and L_Het_ × L_Nod_ scores are significantly different among fibrosis stages. The mean L_Het_, L_Nod_, and L_Het_ × L_Nod_ scores in severe fibrosis stages F2 and F3 were significantly higher than those in mild fibrosis F1 (as shown in Table 2). Thus, it is notable that quantified L_Het_, L_Nod_, and L_Het_ × L_Nod_ scores can provide information for diagnosing hepatic fibrosis. In previous studies, several imaging methods were reported for differentiating hepatic fibrosis and cirrhosis. A study [25] compared the diagnostic accuracy between gadoxetic acid-enhanced MR imaging, transient elastography, and ultrasound shear wave elastography point quantification (ElastPQ). The gadoxetic acid-enhancement index showed similar diagnostic accuracy for significant fibrosis (≥ F2) or cirrhosis (F4) when using transient elastography (AUROC 0.866 and 0.884) or ElastPQ (AUROC 0.751 and 0.786), respectively. A comparative study [26] using hepato-biliary phase imaging (relative enhancement), susceptibility-weighted imaging (SWI; liver-to-muscle ratio), and diffusion-weighted imaging (DWI; apparent diffusion coefficient: ADC value) reported that the AUROC of SWI showed higher value for diagnosing cirrhosis (F4) than the hepato-billiary phase image and DWI (0.92 vs. 0.80 and 0.79), and the AUROC of the combination of all of these showed the highest value for diagnosing cirrhosis (0.93). A recent study [27] of gadoxetic acid-enhanced MR imaging using a radiomics model based on texture analysis reported that the AUROCs of the radiomic fibrosis index were 0.90, 0.89, and 0.91 for significant fibrosis, advanced fibrosis, and cirrhosis, respectively. In the present study, L_Het_ and L_Nod_ scores have similar, excellent diagnostic accuracy for significant fibrosis (≥ F2). Thus, the L_Het_ and L_Nod_ quantification can be a noninvasive technique capable of detecting fibrotic changes within the liver parenchyma in CLD. The major strengths of the integrated WALTS program include the ability to evaluate previously obtained liver MR or CT images (useful for retrospective large-scale population studies), wide availability of MR and CT imaging, no requirement for intravenous contrast media injection, and no additional hardware requirements for image acquisition procedures. Moreover, the L_Het_ and L_Nod_ quantification program may help predict cirrhosis, liver compensation, and death [16]. Therefore, this MRI-compatible WALTS software may be useful for clinical application to various liver diseases including CLD.

Diagnostic accuracy, reproducibility, and repeatability in L_Het_ and L_Nod_ measurements are crucial for assessing the diagnostic performance of an imaging technique [28]. The L_Het_ and L_Nod_ scores derived from routine liver MR images showed good reproducibility between two different observers in diagnosing hepatic fibrosis. WALTS software can quantify axial 3D-THRIVE MR images in less than 5 min. The applicability of WALTS to retrospective clinical studies has great merits since it allows us to predict liver fibrosis and compare disease progression during prospective long-term follow-up studies. 

This study included several limitations. First, this study is a retrospective study with relatively small population size and dealt with CLD patients with heterogeneous underlying disease causes as given in Table 1. The patient cohort used largely included hepatitis B and C. However, there was no consideration of the heterogeneous disease causes in enrolled subjects. Thus, the imaging findings and the predictive power in a larger cohort might be diverse in the patients without viral hepatitis. Also, the distribution skewness (the F0 group is 16/127 patients (12.6% in study population) and F1 group is only 9/127 patients (<10%)) might cause the potentially skewed results; it can lead to spectrum bias in the study population. Future study is needed for a validation study for strengthening the translational impacts in another larger cohort with even subgroups. Although we included a pathologically confirmed F0 group for comparison, this study used the only FIB-4 index to stage liver fibrosis in CLD as a standard for comparison. Further research would be useful to directly compare our method to FibroScan, which currently represents the most utilized technology in the evaluation of fibrosis. FIB-4 index is good for distinguishing cirrhosis from lower fibrosis stages and even then has a modest accuracy. However, this index might potentially lead to false-positive or true-negative findings due to moderate discrimination accuracy and its own limitations. Second, this study did not consider the relationship between L_Het_ and L_Nod_ scores and complications of liver fibrosis. A Lee et al. study [10] reported that quantified L_Het_ scores are correlated with serologic indices, reflecting liver functional status. Smith et al. [13] reported that a single L_Nod_ score allows the prediction of decompensated cirrhosis and death. Sartoris et al. [29] reported that portal hypertension can be detected using a CT-based L_Nod_ score with a high degree of reliability. Considering these findings, future studies are needed to investigate the correlation between L_Het_ and L_Nod_ scores and complications of liver fibrosis. Third, this study performed the reproducibility test in the L_Het_ and L_Nod_ measurements at a single center. Although the findings showed good reproducibility, the L_Het_ score could be influenced by image noise. Further studies are needed for external or cross-validation of diverse datasets with a large-scale cohort across modality, vendors, study protocols, and external validation at multiple centers. Fourth, this study was only focused on patients with hepatic fibrosis (F0–F3). Therefore, further study is needed to clarify the finding in which patients have liver fibrosis including liver cirrhosis (F4) for actual clinical settings and practices. Also, the future development of a prospective study with a larger population size which incorporates cirrhotic patients with advanced fibrosis stage (F4) may offer further insights on how this new methodology may expand the current standard of care.

## 5. Conclusions

This study developed an integrated semiautomatic software for the quantification of hepatic heterogeneity and nodularity, and the measurements of L_Het_ and L_Nod_ scores are reproducible in assessing fibrosis stage in CLD. The combination of quantitative L_Het_ and L_Nod_ scores may be more useful for differentially diagnosing the fibrosis stage in CLD using routine MR images.

## Figures and Tables

**Figure 1 jcm-10-01697-f001:**
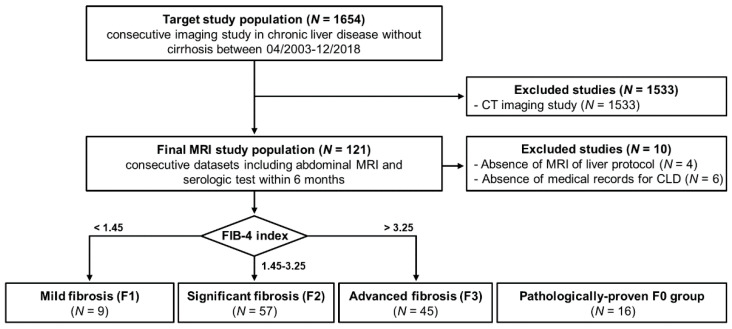
Flowchart for inclusion of chronic liver disease. FIB-4 index = the serum biomarkers of fibrosis-4 index.

**Figure 2 jcm-10-01697-f002:**
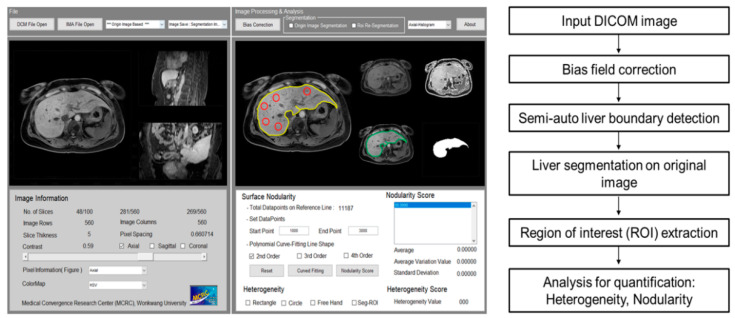
Graphical user interface (GUI) of a tailor-made quantification software for assessing liver heterogeneity (L_Het_) and nodularity (L_Nod_) (**left** side) and flowchart for quantifying liver fibrosis (**right** side).

**Figure 3 jcm-10-01697-f003:**
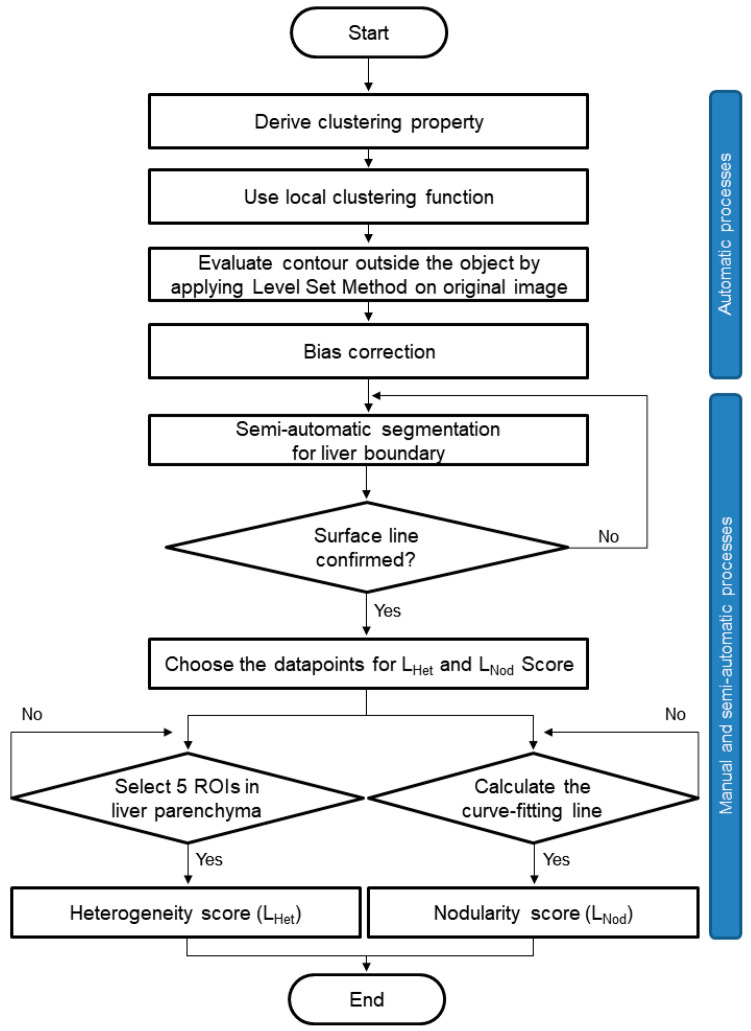
Overall image postprocessing procedures for hepatic heterogeneity (L_Het_) and nodularity (L_Nod_) quantification using MR images.

**Figure 4 jcm-10-01697-f004:**
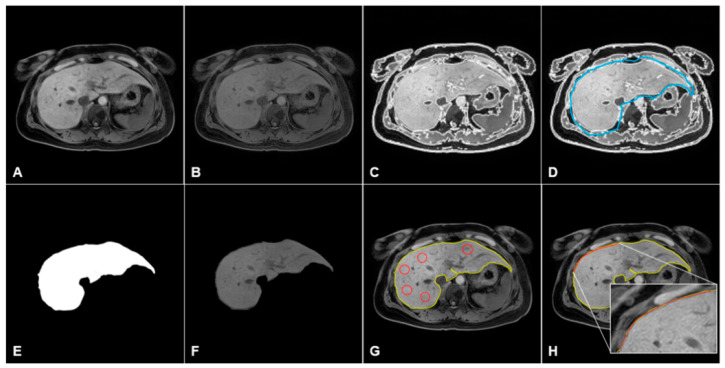
Representative postprocessing images for liver heterogeneity (L_Het_) and nodularity (L_Nod_) quantification in a CLD patient: (**A**) raw MRI data, (**B**) bias field correction, (**C**) liver contour detection, (**D**) drawn region of interest (ROI, blue line), (**E**) segmented liver binary image, (**F**) liver ROI extraction, (**G**) L_Het_ and L_Nod_ quantification (L_Het_ ROIs = 5 red circles; liver surface line for L_Nod_ = yellow line), and (**H**) curve-fitting line (red line) for L_Nod_ quantification along the liver surface line.

**Figure 5 jcm-10-01697-f005:**
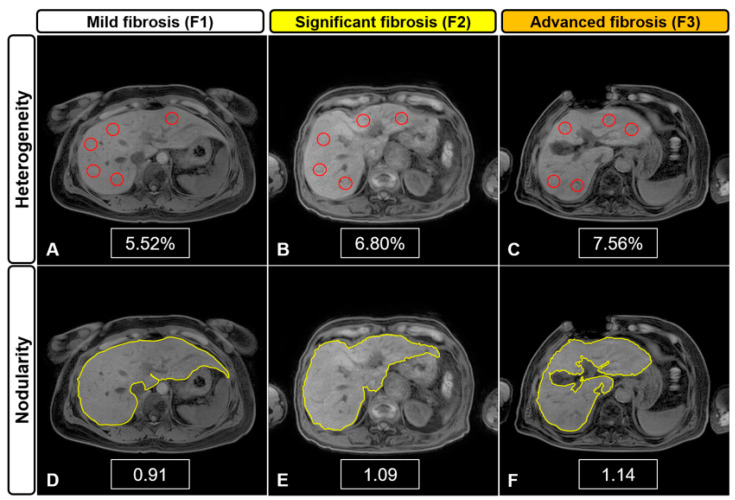
Representative MR images for liver heterogeneity (L_Het_) and nodularity (L_Nod_) quantification according to Figure 1. F1 (**A**,**D**), F2 (**B**,**E**), and F3 (**C**,**F**). Higher heterogeneity and nodularity scores are seen with increased severity in the fibrotic liver.

**Figure 6 jcm-10-01697-f006:**
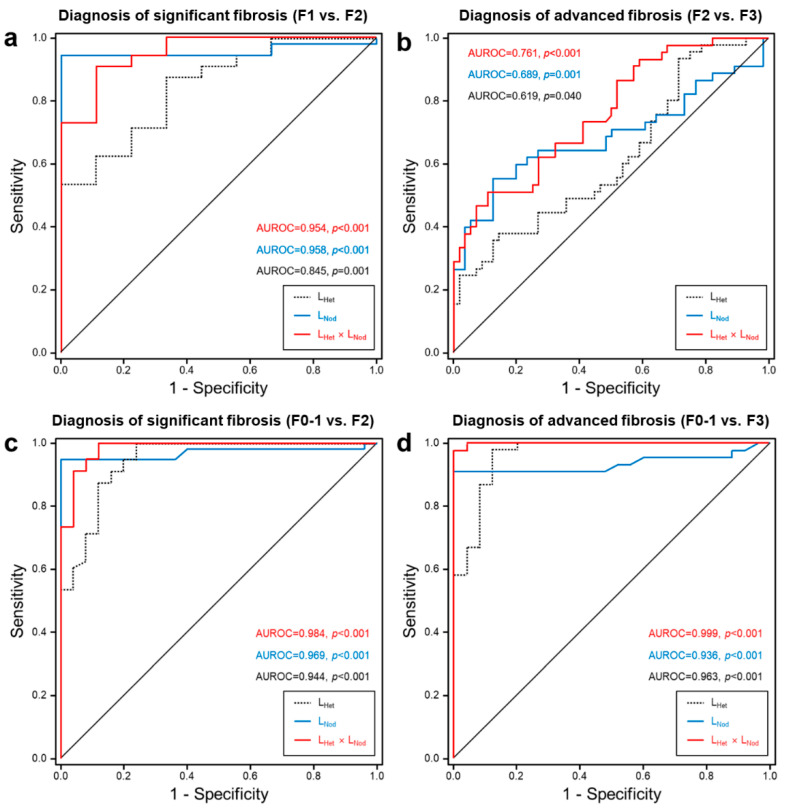
AUROC curves of liver heterogeneity (L_Het_, dotted line), nodularity (L_Nod_, blue line), and combined L_Het_ × L_Nod_ (red line) scores for the differentiation of fibrosis stages: (**a**) diagnosis of significant fibrosis (F1 vs. F2), (**b**) diagnosis of advanced fibrosis (F2 vs. F3), (**c**) diagnosis of significant fibrosis (combined F0–1 vs. F2), and (**d**) diagnosis of advanced fibrosis (combined F0–1 vs. F3). AUROC = area under the receiver operating characteristic curve.

**Table 1 jcm-10-01697-t001:** Etiology of study population and clinical data of serum biochemistry in subgroups of chronic liver disease.

	F0 (*n* = 16)	F1 (*n* = 9)	F2 (*n* = 57)	F3 (*n* = 45)	*p*-Value * ^†^	Multiple Comparisons		
						^a^ F1 vs. F2	^b^ F1 vs. F3	^c^ F2 vs. F3
Etiology *								
Unknown	16 (100)	-	-	-				
Hepatitis B	-	6 (67)	37 (65)	27 (60)	* 0.570			
Hepatitis C	-	0 (0)	5 (9)	4 (9)				
Coinfection	-	0 (0)	2 (4)	0 (0)				
Alcohol	-	1 (11)	5 (9)	6 (13)				
NAFLD	-	0 (0)	6 (10)	5 (11)				
Autoimmune	-	2 (22)	2 (3)	3 (7)				
Serum biochemistry ^†^								
Age (year)	35.0 ± 15.5	50.3 ± 14.9	60.3 ± 12.1	64.8 ± 13.6	^†^ 0.008 ^b^	0.086	0.008	0.191
BMI	24.8 ± 3.2	24.8 ± 1.7	24.6 ± 2.0	25.3 ± 2.9	^†^ 0.211	0.953	0.777	0.184
Albumin (g/dL)	4.36 ± 0.07	3.80 ± 0.31	4.08 ± 0.43	4.30 ± 0.69	^†^ 0.873	0.962	0.888	0.928
ALP (IU/L)	73.9 ± 8.3	279.2 ± 36.7	285.0 ± 17.9	459.8 ± 53.6	^†^ 0.002 ^c^	0.998	0.125	0.002
ALT (IU/L)	18.8 ± 2.1	31.7 ± 8.7	37.5 ± 3.8	74.2 ± 25.6	^†^ 0.224	0.989	0.552	0.230
AST (IU/L)	21.0 ± 1.4	26.0 ± 4.4	38.7 ± 3.6	109.1 ± 36.0	^†^ 0.058	0.972	0.311	0.064
Bilirubin (mg/dL)	0.64 ± 0.09	0.73 ± 0.09	0.78 ± 0.09	2.66 ± 0.90	^†^ 0.058	1.000	0.441	0.052
GGT (IU/L)	38.5 ± 10.9	83.4 ± 29.9	63.2 ± 7.2	184.9 ± 30.0	^†^ 0.001 ^c^	0.946	0.266	<0.001
Platelet count (10^3^/μL)	243.5 ± 8.2	223.9 ± 17.4	172.3 ± 7.0	135.4 ± 6.9	<^†^ 0.001 ^a b c^	0.014	<0.001	0.001

Groups: F1, mild fibrosis group; F2, significant fibrosis group; and F3, advanced fibrosis group. Abbreviations: ALP: alkaline phosphatase; ALT: alanine aminotransferase; AST: aspartate aminotransferase; GGT: gamma-glutamyl transpeptidase; NAFLD; nonalcoholic fatty liver disease. Etiology data are presented as the number of patients. The value in parenthesis indicates the percentage as the number of patients/total number of patients ×100. Serum biochemistry data are presented as mean ± SEM. Reference ranges for serum biochemistry are as follows: 3.4–5.4 g/dL for albumin, 44–147 IU/L for ALP, 0–35 IU/L for ALT, 0–35 IU/L for AST, 0.1–1.2 mg/dL for total bilirubin, 8–38 IU/L for GGT, and 120–473 × 10^3^/μL for platelet count. * The difference among fibrosis groups in etiology data was analyzed by Pearson’s chi-square test. ^†^ The difference among three fibrosis groups was analyzed by one-way ANOVA with Tukey post hoc test as follows: ^a^ F1 vs. F2, ^b^ F1 vs. F3, and ^c^ F2 vs. F3.

**Table 2 jcm-10-01697-t002:** Comparison of liver heterogeneity (L_Het_) and nodularity (L_Nod_) scores according to fibrosis stages (F).

	F0 (*n* = 16)	F1 (*n* = 9)	F2 (*n* = 57)	F3 (*n* = 45)	*p*-Value *	Multiple Comparisons		
						^a^ F1 vs. F2	^b^ F1 vs. F3	^c^ F2 vs. F3
Heterogeneity (L_Het_, %)	3.49 ± 0.34	5.52 ± 0.88	6.74 ± 0.89	7.56 ± 1.79	<0.001^a b c^	0.032	<0.001	0.008
Nodularity (L_Nod_)	0.84 ± 0.06	0.91 ± 0.04	1.09 ± 0.08	1.14 ± 0.14	<0.001 ^a b c^	<0.001	<0.001	0.022
L_Het_ × L_Nod_ score	2.96 ± 0.46	5.01 ± 0.91	7.30 ± 0.89	8.48 ± 1.34	<0.001 ^a b c^	<0.001	<0.001	<0.001

Data are presented as mean ± SD. The final L_Het_ and L_Nod_ scores in each fibrosis group were calculated as an averaged score obtained by reporting scores of two observers (observer A: Y.R.K., observer B: Y.H.L.) for AUROC differential diagnosis according to fibrosis stages. The average scores in F0 group were used as the reference ranges. * The difference among the three fibrosis groups was analyzed by one-way ANOVA with Tukey post hoc test as follows: ^a^ F1 vs. F2, ^b^ F1 vs. F3, and ^c^ F2 vs. F3.

**Table 3 jcm-10-01697-t003:** Receiver operator curve analysis for diagnosing fibrosis stage using liver heterogeneity (L_Het_) and nodularity (L_Nod_) scores.

Comparison	Threshold Value	Sensitivity (%)	Specificity (%)	PPV (%)	NPV (%)	DA (%)	AUROC	*p*-Value
*L_Het_ score*								
F1 vs. F2	6.22	71.4 (41/57)	77.8 (7/9)	95.3 (41/43)	30.4 (7/23)	72.7 (48/66)	0.845	0.001
F2 vs. F3	7.02	53.3 (24/45)	53.6 (30/57)	47.1 (24/51)	58.8 (30/51)	52.9 (54/102)	0.619	0.040
F0–1 vs. F2	5.68	87.5 (50/57)	88.0 (22/25)	94.3 (50/53)	75.9 (22/29)	87.8 (72/82)	0.944	<0.001
F0–1 vs. F3	6.17	86.7 (39/45)	88.0 (22/25)	92.9 (39/42)	78.6 (22/28)	87.1 (61/70)	0.963	<0.001
*L_Nod_ score*								
F1 vs. F2	0.97	94.6 (54/57)	100.0 (9/9)	100.0 (54/54)	75.0 (9/12)	95.4 (63/66)	0.958	<0.001
F2 vs. F3	1.12	64.4 (29/45)	64.9 (37/57)	59.2 (29/49)	69.8 (37/53)	64.7 (66/102)	0.689	0.001
F0–1 vs. F2	0.95	94.6 (54/57)	96.0 (24/25)	98.2 (54/55)	88.9 (24/27)	95.1 (78/82)	0.969	<0.001
F0–1 vs. F3	0.94	91.1 (41/45)	92.0 (23/25)	95.3 (41/43)	85.2 (23/27)	91.4 (64/70)	0.936	<0.001
*L_Het_ × L_Nod_ score*								
F1 vs. F2	6.10	89.5 (51/57)	88.9 (8/9)	98.1 (51/52)	57.1 (8/14)	89.4 (59/66)	0.954	<0.001
F2 vs. F3	7.85	66.7 (30/45)	68.4 (39/57)	62.5 (30/48)	72.2 (39/54)	67.6 (69/102)	0.761	<0.001
F0–1 vs. F2	5.98	91.1 (52/57)	92.0 (23/25)	96.3 (52/54)	82.1 (23/28)	91.5 (75/82)	0.984	<0.001
F0–1 vs. F3	6.70	97.8 (44/45)	100.0 (25/25)	100.0 (44/44)	96.2 (25/26)	98.6 (69/70)	0.999	<0.001

Note—Data in parentheses are raw data used to calculate percentages. AUROC: area under the receiver operator curve; DA: diagnostic accuracy = (TP + TN)/(TP + FP + TN + FN); F: fibrosis stages; FN: false negative; FP: false positive; NPV: negative predictive value; PPV: positive predictive value; TN: true negative; TP: true positive.

**Table 4 jcm-10-01697-t004:** Intra- and interobserver variability in liver heterogeneity (L_Het_) and nodularity (L_Nod_) measurements according to fibrosis stages.

	Observer A	Observer B	*p*-Value *	Intrarater Reliability (ICC) ^†^	95% CI		*p*-Value ^†^
					Lower Bound	Upper Bound	
*L_Het_* *score*							
Overall (*n* = 111)	7.06 ± 1.68 (24)	6.94 ± 1.64 (24)	0.140	0.718	0.589	0.806	<0.001
F1 (*n* = 9)	5.45 ± 1.08 (20)	5.58 ± 0.85 (15)	0640	0.782	0.035	0.951	0.023
F2 (*n* = 57)	6.85 ± 1.15 (17)	6.75 ± 1.12 (13)	0.535	0.624	0.362	0.779	<0.001
F3 (*n* = 45)	7.65 ± 2.05 (27)	7.47 ± 2.08 (28)	0.558	0.679	0.417	0.824	<0.001
*L_Nod_* *score*							
Overall (*n* = 111)	1.10 ± 0.12 (11)	1.10 ± 0.14 (13)	0.459	0.832	0.755	0.885	<0.001
F1 (*n* = 9)	0.91 ± 0.04 (4)	0.90 ± 0.06 (7)	0.869	0.768	−0.029	0.948	0.027
F2 (*n* = 57)	1.09 ± 0.09 (8)	1.09 ± 0.10 (9)	0.905	0.601	0.322	0.765	<0.001
F3 (*n* = 45)	1.14 ± 0.13 (11)	1.15 ± 0.16 (14)	0.596	0.852	0.731	0.919	<0.001

Abbreviations: ICC: intraclass correlation coefficient; CI: confidence interval. L_Het_ and L_Nod_ scores of each observer are presented as means ± SD (mean coefficient of variance, %). * The differences between both observers in L_Het_ and L_Nod_ scores were assessed by the paired *t*-test. ^†^ The intrarater reliability between both observers was assessed by the intraclass correlation (ICC) test.

## Data Availability

All anonymized data sources described in this study are available from the corresponding author on reasonable request.

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
