# Peer review of "Assessment of Liver Fibrosis Stage Using Integrative Analysis of Hepatic Heterogeneity and Nodularity in Routine MRI with FIB-4 Index as Reference Standard"

_jcm, 2021, doi:10.3390/jcm10081697_

Round 1

Reviewer 1 Report

Dear Authors,

The methodology presented is interesting and useful in the evaluation and quantification of fibrosis in patients with chronic liver disease. Yet, these are some limitations:

1)         this study only utilizes the FIB-4 as a standard for comparison, yet it would be useful to directly compare this method to Fibroscan which currently represents the most utilized in technology in the evaluation of fibrosis; it would be useful if a direct comparison with data obtained via Fibroscan could also be included.

2)         BMI of the patients should be included as part of the clinical data in table 1. It would be important to see whether this technology offers any advantages in the quantification of fibrosis in obese patients when compared to the use of Fibroscan. This currently still remains a shortcoming of Fibroscan due its relatively lower accuracy in this subset of patients.

3)         This is a retrospective study with a relatively small population size; the future development of a prospective study with a larger population size which also incorporates cirrhotic patients with advanced fibrosis stage (F4) may offer further insights on how this new methodology may expand the current standard of care.

We recognize that it may be technically difficult to address point 3 at the present time; yet, trying to address points one 1 and 2 may be helpful to further assess the added benefit of the proposed technology.

Overall, we do recognize the novelty of this study and the benefits that it may offer.

Author Response

The methodology presented is interesting and useful in the evaluation and quantification of fibrosis in patients with chronic liver disease. Yet, these are some limitations:

→ Thank you for your kind review.

1) this study only utilizes the FIB-4 as a standard for comparison, yet it would be useful to directly compare this method to Fibroscan which currently represents the most utilized in technology in the evaluation of fibrosis; it would be useful if a direct comparison with data obtained via Fibroscan could also be included.

→ As your suggestion, we have check Fibroscan or elastography data. Unfortunately, our retrospective research (WKUH-2017-03-026) has already finished and collected dataset were not included the Fibroscan data. Thus, we have included as a limitation and further study about above issue. Please see the Lines 22-25 in DISCUSSION section on Page 12.

2) BMI of the patients should be included as part of the clinical data in table 1. It would be important to see whether this technology offers any advantages in the quantification of fibrosis in obese patients when compared to the use of Fibroscan. This currently still remains a shortcoming of Fibroscan due its relatively lower accuracy in this subset of patients.

→ As your comment, we have added the BMI of patients in Table 1 on Page 7.

3) This is a retrospective study with a relatively small population size; the future development of a prospective study with a larger population size which also incorporates cirrhotic patients with advanced fibrosis stage (F4) may offer further insights on how this new methodology may expand the current standard of care.

→ About above issue, we have included the several sentences as an limitation. Please see the Lines 22-25 on Page 12 and the Lines 22-25 on Page 13 in the DISCUSSION section.

We recognize that it may be technically difficult to address point 3 at the present time; yet, trying to address points one 1 and 2 may be helpful to further assess the added benefit of the proposed technology.

Overall, we do recognize the novelty of this study and the benefits that it may offer.

→ Thank you for your kind review.

Reviewer 2 Report

Kim et al. have submitted a study for publication that details a novel approach to assessing fibrosis via non-invasive means as a computational method in MRI.  The study is appropriate for translational impact and represents an appropriate step forward in the diagnosis of the stage of fibrosis.  The study is well-written and demonstrates the utility of the computational approach.  

I do think an issue must be addressed by the authors both in data presentation as well as in the text.  The patient cohort used is heavily skewed to Hepatitis B and Hepatitis C.  Although the data and method suggest a powerful tool for liver fibrosis regardless of etiology, it could be interpreted differently.  That is to say that the majority of the results and the predictive power of the tool might be questionable in a larger cohort that is more diverse or without viral hepatitis patients.  

The authors should increase their study inclusion and/or validate their findings in another cohort to strengthen the translational impact of these findings and include.  In addition, this should be addressed in the discussion.  

Author Response

Kim et al. have submitted a study for publication that details a novel approach to assessing fibrosis via non-invasive means as a computational method in MRI. The study is appropriate for translational impact and represents an appropriate step forward in the diagnosis of the stage of fibrosis. The study is well-written and demonstrates the utility of the computational approach.

→ Thank you for your kind review.

I do think an issue must be addressed by the authors both in data presentation as well as in the text. The patient cohort used is heavily skewed to Hepatitis B and Hepatitis C. Although the data and method suggest a powerful tool for liver fibrosis regardless of etiology, it could be interpreted differently. That is to say that the majority of the results and the predictive power of the tool might be questionable in a larger cohort that is more diverse or without viral hepatitis patients. The authors should increase their study inclusion and/or validate their findings in another cohort to strengthen the translational impact of these findings and include.  In addition, this should be addressed in the discussion.

→ About above issue, we have addressed the limitation in enrolled patient cohort. Additionally, we have performed the statistical analysis in comparisons with combined F0-F1 group (n=25) vs. F2 group (n=57) and F0-F1 group (n=25) vs. F3 group (n=45). Please see the Lines 22-25 and the Lines 22-25 on Page 12 in the DISCUSSION section; and Table 3.

Reviewer 3 Report

In this study the authors performed assessment of liver fibrosis stage using integrative analysis of hepatic heterogeneity and nodularity in routine MRI.

The manuscript is written well and reads well. However the manuscript has several shortcomings

  1. The approach mentioned is not entirely novel. Its a combination of the two methods
  2. The reference standard for fibrosis is not gold standard-liver biopsy. FIB-4 is good for distinguishing cirrhosis from lower fibrosis stages and even then has a modest accuracy. So the reference standard is a has its own limitations
  3. Spectrum bias in the study population. The F1 group is only 9/111 patients which is <10% of the population making the results skewed to high accuracy. Perhaps the real accuracies are the one distinguishing between F2 and F3 ~0.70

Specific comments

  1. Surprised to see that the healthy control group had a liver biopsy. This is usually not possible in many high quality studies. Not sure how the authors managed to get these controls undergo liver biopsy. Or these subjects were liver donors? If so please clarify

  2. Mention the accuracy of FIB-4 in distinguishing the stages
  3. Expand WALTS-
  4. If you had segmented entire liver why was the entire liver heterogeneity was taken into consideration. perhaps that would have been better discriminator
  5. Did the readers chose the same slice. This may account for high inter-observer agreement. 
  6. How did the readers chose the slice for analysis?
  7. For F1 vs.F2 analysis did you combine Normal + F1 as one group and F2+F3 as the second group? Its not clear from the statistical description. Same for other groups
  8. Abstract mentions F0- No other description of this category in the text

Agree that the measurements can be reproducible in clinical MR images, however the interpretation of the accuracy needs clarification.

They did not compare or at least discuss with the current  most accurate test MR Elastography which has ICC >95% and accuracy >90%. The Lhet and Lnod has ICC of only 0.72 and 0.83 which is quite low compared to MRE.

Author Response

In this study the authors performed assessment of liver fibrosis stage using integrative analysis of hepatic heterogeneity and nodularity in routine MRI. The manuscript is written well and reads well. However the manuscript has several shortcomings.

  1. The approach mentioned is not entirely novel. Its a combination of the two methods

→ As you mentioned above, we have focused on the imaging-based assessment using only routine T1 MR images. As your understanding (last your comment), current MR elastography (MRE) method is accurate and superior to routine MR imaging for the non-invasive diagnosis of significant liver fibrosis and cirrhosis. Also, without the help of exogenous MRI contrast agents, the diagnostic accuracy of liver fibrosis through the quantification of routine T1 MR images is lower than that of MRE [1]. However, the MRE technique for evaluating accurate liver stiffness is required the standardized acquisition technique and radiologist’s expertness.

As we mentioned in the INTRODUCTION section, recent studies used a single measurement, either liver heterogeneity (LHet) or nodularity (LNod), for computer-based evaluating liver fibrosis; the method was not clearly distinguished among fibrosis subgrades. Up to date, it is difficult challenge in clinical.

In the present study, by using only routine T1 MRI, developed integrative (LHet, LNod) software and analysis processes are helpful for differential diagnosis of fibrosis subgroups in chronic liver disease. Although this study included limitations and shortcomings, please reconsider our advances in clinical practice and setting.

[1] Non-invasive detection of liver fibrosis: MR imaging features vs. MR elastography. Abdom Imaging. 2015 Apr; 40(4): 766–775.

  1. The reference standard for fibrosis is not gold standard-liver biopsy. FIB-4 is good for distinguishing cirrhosis from lower fibrosis stages and even then has a modest accuracy. So the reference standard is a has its own limitations

→ We have included several sentences with references in the INTRODUCTION and DISCUSSION sections about this issue. Please see the Lines 5-12 on Page 1 and the Lines 30-38 on Page 12.

  1. Spectrum bias in the study population. The F1 group is only 9/111 patients which is <10% of the population making the results skewed to high accuracy. Perhaps the real accuracies are the one distinguishing between F2 and F3 ~0.70

→ Additionally, we have performed the statistical analysis in comparisons with combined F0-F1 group (n=25) vs. F2 group (n=57) and F0-F1 group (n=25) vs. F3 group (n=45). Please see the Figure 6 on Page 9 and Table 3 on Page 10. Also, several results and sentences have included in the RESULTS and DISCUSSION sections. Please see the Lines 1-24 in ‘Sub-heading 3.3’ of RESULTS section on Page 8 and the Lines 26-30 in DISCUSSION section on Page 12.

Specific comments

  1. Surprised to see that the healthy control group had a liver biopsy. This is usually not possible in many high quality studies. Not sure how the authors managed to get these controls undergo liver biopsy. Or these subjects were liver donors? If so please clarify

→ We have described how control group undergo liver biopsy. Please see the Lines 4-7 in the METHOD section on Page 3.

  1. Mention the accuracy of FIB-4 in distinguishing the stages

→ To distinguish the fibrosis stages, this study utilized the FIB-4 index as a standard for comparison. We have added the diagnostic accuracy (DA, %) in discriminating the FIB-4 stages. Please see the results in Table 3 on Page 10; the Lines 25-29 on Page 8 and Lines 1-2 on Page 9 in ‘Sub-heading 3.3’ of RESULTS section.

  1. Expand WALTS-

→ We have included the expansion of WALTS. Please see the Lines 2-4 in ‘Sub-heading 2.4’ of METHOD section on Page 3.

  1. If you had segmented entire liver why was the entire liver heterogeneity was taken into consideration. perhaps that would have been better discriminator

→ We have included more detailed description about above issue. Please see the Lines 7-13 on Page 6.

  1. Did the readers chose the same slice. This may account for high inter-observer agreement.

→ We have included more detailed description how to choose the same slice for analysis. Please see the Lines 7-13 and Lines 17-21 on Page 6.

  1. How did the readers chose the slice for analysis?

→ We have included more detailed description how to choose the same slice for analysis. Please see the Lines 7-13 and Lines 17-21 on Page 6.

  1. For F1 vs.F2 analysis did you combine Normal + F1 as one group and F2+F3 as the second group? Its not clear from the statistical description. Same for other groups

→ We have included the findings in comparisons of “F1 vs.F2,” and “F2 vs.F3” as well as additionally “combined F0-F1 group vs. F2 group” and “combined F0-F1 group vs. F3 group.” Please see the Figure 6 (Page 9), Table 3 (Page 10) and the Lines 1-24 in ‘Sub-heading 3.3’ of RESULTS section on Page 8.

  1. Abstract mentions F0- No other description of this category in the text

→ We have modified in the text for this issue.

Agree that the measurements can be reproducible in clinical MR images, however the interpretation of the accuracy needs clarification.

→ To clarify the measurements for intra- and inter-observer agreement, we have included several sentences in METHOD section. Please see the Lines 17-21 on Page 6.

They did not compare or at least discuss with the current most accurate test MR Elastography which has ICC >95% and accuracy >90%. The LHet and LNod has ICC of only 0.72 and 0.83 which is quite low compared to MRE.

→ We have included the several sentences with a new reference in the DISCUSSION section. Please see the Lines 23-27 on Page 11.

Round 2

Reviewer 2 Report

None.  Thank you for addressing the points I raised in my previous review.  

Author Response

None.  Thank you for addressing the points I raised in my previous review.

→ Thank you so much for your kind review.

Reviewer 3 Report

There are still major issues with the manuscript

  1. Title needs to have FIB-4 Index as reference standard
    1. Assessment of liver fibrosis stage using integrative analysis of hepatic heterogeneity and nodularity in routine MRI with FIB-4 Index as reference standard
  2. Need English language editing for the entire manuscript
  3. The authors mention that the “healthy controls “presented to hospital and had symptoms of fatigue and malaise and clinically were suspected to have liver disease. Presuming that they had abnormal liver function tests which prompted a liver biopsy. So, by any standard definition these are not healthy controls. The use of term healthy control for this group of individuals is inappropriate and misleading. So, the entire manuscript needs to readdress this as F0 individuals. Put the cause of chronic liver disease as unknown.
  4. Abstract
    1. Fibrosis is given stages and not grades-change it. Inflammation is graded. Use staging consistently throughout the manuscript
    2. Patient population -111 CLD patients and 16 subjects with suspected liver disease who underwent liver biopsy
  5. Introduction has abrupt changes of thoughts and poorly written. The reasoning for page 1 line 14 does not follow well after the information about Fib-4 index. There are no sentences prior to justify how non-invasive diagnosis of hepatic fibrosis playing a critical role in control of liver disease progression and in therapeutic judgement.
  6. Page 6- avoid terms like “pure”
  7. Provide the normal cut-off values for Albumin, ALP, ALT, AST, Bilirubin etc. There can be provided in the footnote. Please don’t middle justify the foot notes for all the tables. Justify to the left.

Author Response

Reviewer 3

There are still major issues with the manuscript

  1. Title needs to have FIB-4 Index as reference standard

Assessment of liver fibrosis stage using integrative analysis of hepatic heterogeneity and nodularity in routine MRI with FIB-4 Index as reference standard

→ As your comment, we have included words “with FIB-4 Index as reference standard” in the title. Please see the title on Page 1.

  1. Need English language editing for the entire manuscript

→ Previously, we have utilized the English editing service through AJE website. We have attached the certification as a Figure. At this time, we unfortunately have to submit the revised manuscript within only 3 days and have edited the revised manuscript for only present issues (from Prof. S. Thirunavukkarasu in Annamalai University).

  1. The authors mention that the “healthy controls “presented to hospital and had symptoms of fatigue and malaise and clinically were suspected to have liver disease. Presuming that they had abnormal liver function tests which prompted a liver biopsy. So, by any standard definition these are not healthy controls. The use of term healthy control for this group of individuals is inappropriate and misleading. So, the entire manuscript needs to readdress this as F0 individuals.

→ We have re-addressed the term “healthy controls” to F0 individuals in the text for above issue.

Put the cause of chronic liver disease as unknown.

→ In Table 1 on Page 7, Put the cause of chronic liver disease as unknown.

  1. Abstract

Fibrosis is given stages and not grades-change it. Inflammation is graded. Use staging consistently throughout the manuscript

→ Throughout the manuscript, we have changed the use of word “grade”. Please see the abstract and manuscript.

Patient population -111 CLD patients and 16 subjects with suspected liver disease who underwent liver biopsy

→ We have changed the sentence as your suggestions.

  1. Introduction has abrupt changes of thoughts and poorly written. The reasoning for page 1 line 14 does not follow well after the information about Fib-4 index. There are no sentences prior to justify how non-invasive diagnosis of hepatic fibrosis playing a critical role in control of liver disease progression and in therapeutic judgement.

→ We have modified the sentences in the INTRODUCTION. Please see the first paragraph on Pages 1-2.

  1. Page 6- avoid terms like “pure”

→ We have eliminate the term.

  1. Provide the normal cut-off values for Albumin, ALP, ALT, AST, Bilirubin etc. There can be provided in the footnote.

→ We have included the normal cut-off values for serum chemistry levels in the footnote.

Please don’t middle justify the foot notes for all the tables. Justify to the left.

→ For 1st submission, we have prepared the foot notes to the left for all the tables. Before review, our manuscript changed to JCM format. As my understanding JCM format, the table footnote format in JCM is middle. For this revision, we have prepared again to the left. Please take our circumstances into reconsideration.
